# Intestinal Microbiome in Dogs with Chronic Hepatobiliary Disease: Can We Talk about the Gut–Liver Axis?

**DOI:** 10.3390/ani13203174

**Published:** 2023-10-11

**Authors:** Verena Habermaass, Daniela Olivero, Eleonora Gori, Chiara Mariti, Erika Longhi, Veronica Marchetti

**Affiliations:** 1Department of Veterinary Sciences, University of Pisa, Via Livornese Lato Monte, 56122 Pisa, Italy; verena.habermaass@phd.unipi.it (V.H.); eleonora.gori@unipi.it (E.G.); veronica.marchetti@unipi.it (V.M.); 2Analysis Lab BSA Scilvet, Via A. D’Aosta 7, 20129 Milan, Italy; 3Analysis Lab Labospace, Via Apelle 41, 20128 Milan, Italy; erika.longhi@labospace.com

**Keywords:** canine microbiota, gut microbiota, dysbiosis, hepatopathy, canine hepatic disease, liver–gut axis

## Abstract

**Simple Summary:**

The interaction between gut and liver is currently considered for the management of humans with chronic hepatobiliary disease (CHD). The gut microbiota (GM) is considered to strongly mediate this crosstalk. The present study aimed to evaluate the GM in dogs diagnosed with CHD. Comparison among CHD dogs were made with respect to some clinical and biochemical variables. Sixty-five dogs were prospectively enrolled. Clinical and hematobiochemical evaluation together with GM sequencing was performed. Several GM modifications were observed, with cholestasis apparently associated with deeper GM alteration. For the management of dogs with CHD and especially cholestatic CHD, clinicians should be aware that gut–liver interaction may lead to dysbiosis.

**Abstract:**

The gut–liver axis represents a current topic in human medicine. Extensive research investigates the gut microbiome (GM) modifications in relation to various kinds of chronic hepatobiliary diseases (CHD), with many mechanisms and therapeutical implications recognized. Those aspects in veterinary medicine are still quite unexplored. The aim of the present study was to evaluate GM in dogs diagnosed with CD. Comparison among CHD dogs were made considering some clinical and biochemical variables (lipemia and alanine–aminotransferase activities), presence of cholestasis or endocrine disorders, diet). Sixty-five dogs were prospectively enrolled with clinical and hematobiochemical evaluation and 16S-RNA GM sequencing assessed. Dogs that received antibiotics and/or pre/pro/symbiotics administration were excluded. Deeper GM alteration was observed between dogs with or without ultrasonographic and biochemical cholestatic CHD. Cholestasis was associated with a decrease in several bacterial taxa, including *Clostridium hiranonis*, *Fusobacterium*, *Megamonas*, *Ruminococcus faecis*, *Turicibacter*, and higher levels of *Escherichia/Shigella* and *Serratia*. Thus, the alteration in bile flow and composition, typical of cholestasis, may directly affect the local intestinal microbial environment. For the management of dogs with CHD and especially cholestatic CHD, clinicians should be aware that gut–liver interaction may lead to dysbiosis.

## 1. Introduction

The term gut microbiome (GM) refers to the whole of microorganisms, including bacteria, inhabiting the gastrointestinal tract, and taking part in a variety of crucial intestinal and extraintestinal functions [1,2,3]. The gut microbial metabolism involves a wide range of molecules ranging from short-chain fatty acids and vitamins to secondary bile acids (BAs) and neurotransmitters [4,5].

In humans, extensive research has investigated the GM modifications in relation to various kinds of liver diseases, and many mechanisms may be involved. Blood from the gut enters the systemic circulation having passed through the liver from the portal vein, meaning that the liver is strategically placed to be involved in the gut microbiome–host interaction [6,7]. Microbiota-derived molecules have the potential to activate the immune system and trigger inflammation [7], providing the liver with the challenging task of balancing tolerance to beneficial and harmless molecules and metabolites with the need to act as a firewall against pathogens and harmful microbe-derived molecules [8]. Disturbances of the gut–liver axis are therefore potentially the key players in several liver diseases.

For example, during cirrhosis and hepatic encephalopathy (HE), ammonia is identified as a key player [9,10,11]. Significant differences have been reported in humans between the GM of cirrhosis or HE patients and healthy control individuals, and some suggest the microbiota alterations in affected patients may contribute to more significant clinical signs of HE via alteration of the intestinal barrier function, an increased proportion of urease-producing bacteria, or in other ways [10,11,12,13].

Dysbiosis in GM was identified also in relation to cholestatic biliary disorders such as primary biliary cholangitis (PBC) and primary sclerosing cholangitis (PSC) [14], and it may be due to the relation between BAs and microbiota; GM is central to bile acid homeostasis and, conversely, bile composition and flow influence gut microbiota [14]. Therefore, it has been postulated that manipulation of the microbiome may contribute to improved outcomes in patients with chronic hepatobiliary disease [15,16]. 

Knowledge on the canine GM is still expanding: novel molecular methods such as DNA sequencing techniques (16S rRNA sequencing) or targeted quantitative polymerase chain reaction (qPCR) assays have allowed for the genetic characterization of the GM in various species [3,17,18]. The main interest in canine GM focus on intestinal disorders [19,20,21,22,23], and the potential interaction between liver disease and GM is still unexplored. One recent study characterized dysbiosis in dogs being medically managed for congenital extrahepatic porto-systemic shunt (CPSS) in relation to diet and antibiotic, lactulose, or probiotic therapy, concluding that dysbiosis appears to be common in dogs medically managed for CPSS, with an unclear clinical significance [24]. The present study aimed to evaluate GM in dogs diagnosed with CHD and, among them, also considering the possible influence of clinical variables as diet, the presence of endocrine disorders, extrahepatic cholestasis, and some clinical–pathological variables.

## 2. Materials and Methods

### 2.1. Animals

Client-owned dogs with a diagnosis of chronic hepatobiliary disease (CHD), of different breed, sex, and weight, referred to the Gastroenterology Service of the Veterinary Teaching Hospital “Mario Modenato” of the University of Pisa, between January 2020 and January 2022, were included. CHD diagnosis was based on the medical history, physical examination, hematology, and blood biochemistry as well as abdominal ultrasonography. Animals with specific persistently increased hepatic enzyme activities (ALT, ALP) and/or biochemical evidence of a reduction in liver function and ultrasonographic evidence of liver parenchymal or biliary abnormalities were classified as having cholestatic or noncholestatic CHD.

Dogs with a history of probiotic/prebiotic/symbiotic or antibiotic administration within 2 months were excluded. At the time of inclusion, dogs had an anamnesis, including information collected about diet and chronic gastrointestinal signs (diarrhea, vomit), and a clinical examination and blood was taken for full analysis. A fresh fecal sample collected by the owner, in a sterile container, was immediately frozen at −20 °C and at −80 °C within the 48 h. When available, information about hepatic histological diagnosis was collected. Dogs with clinical and hematobiochemical suspicious of endocrine disorder were screened with specific hormonal tests (diabetes mellitus, hypothyroidism, hyperadrenocorticism, hypoadrenocorticism) and information about a diagnosis of endocrine disorder was registered.

Serum Alanine Aminotrasnferase (ALT) > 70 U/L was considered increased, whereas hyperlipemia was identified when serum cholesterol was >280 mg/dL (>7.25 mmol/L) and/or triglycerides > 90 mg/dL (>1.02 mmol/L). Diagnosis of cholestatic CHD was based on both blood biochemistry profile (two or more between Alkaline Phosphatase (ALKP) > 250 U/L, Gamma-Glutamyl Transferase (GGT) > 11 U/L, Total Bilirubin > 0.3 mg/dL (>5.13 μmol/L), Cholesterol > 280 mg/dL (>7.25 mmol/L)) and concurrent signs of biliary tract disease at abdominal ultrasound (i.e., biliary sludge or immobile echogenic bile accumulation, abnormalities of the gallbladder wall, enlarged and/or tortuous extrahepatic bile tract).

Microbial taxa were analyzed on stored stool samples, and differences in microbiota composition between dogs that had and had not hyperlipemia/endocrinopathy/increased ALT and also considering diet category, were investigated. The study was conducted in accordance with the Declaration of Helsinki and approved by the Ethics Committee of the University of Pisa (protocol code n. 41, date of approval: 29 October 2020).

### 2.2. Microbiome Analysis

Stool samples consisted of 4 gr of fresh fecal material and were immediately frozen upon collection at −80 °C until processed for DNA extraction. Bacterial DNA was extracted from frozen stool samples with the Stool Cell DNA Extraction Kit (TANBead, W6SCS66) on the Maelstrom 9600 TANBead automated extraction platform following the manufacturer′s instructions. The extracted DNA (6 ng for each sample) was used for microbiome analysis with the NGS method on the IonTorrent platform (Thermo Fisher Scientific, Waltham, MA, USA).

The Ion 16S™ Metagenomics Kit (Thermo Scientific™, Waltham, MA, USA), which analyzes 7 of the 9 hypervariable regions of the bacterial 16S rRNA, was used for the library construction. Libraries were then sequenced on the Ion GeneStudio™ S5 System (Ion Torrent™, Thermo Fisher Scientific, Waltham, MA, USA). Before the analysis, the sequences (or reads) obtained from sequencing were cleaned using dedicated algorithms to remove short and low-quality reads. Sequences of less than 10 were not considered. Sequences generated were directly analyzed using the Ion 16S™ metagenomics analyses module within the Ion Reporter™ v. 5.20.2.0 software using the premium curated Applied Biosystems™ MicroSEQ™ ID 16S rRNA database and curated Greengenes database, enabling a semiquantitative assessment of complex microbial samples. The analysis of the raw sequence data was also carried out by MicroBAT Software that performs the taxonomic assignment of the individual reads by aligning with the RDP database (Ribosomal Database Project). Only the sequences that satisfy certain criteria in the alignment phase are associated by the analysis system at the taxonomic level of the species (evaluation of the minimum length of the sequence that aligns with the reference sequence and percentage of similarity). Using QIIME 2 (Quantitative Insights into Microbial Ecology 2) v. 2019.7, the sequences were processed and analyzed. DADA2 was used to create the amplicon sequence variant (OTUs) table after the sequences were demultiplexed. QIIME2 was used to evaluate alpha diversity using Chao1 (richness), Shannon diversity, and observed ASVs metrics. Beta diversity was estimated using Bray–Curtis, Jaccard, and unweighted UniFrac distance matrices; plots were generated in QIIME2.

### 2.3. Statistical Analysis

Normal distribution was investigated by the Kolmogorov–Smirnov test. Non-normally distributed data were expressed as median and range whereas normally distributed data were expressed as mean ± SD. Comparison of alpha diversity indices (Chao1, Shannon′s diversity, and observed OTUs) between groups was performed dependent on normal distribution using the Mann–Whitney test and compared with each other. The Mann–Whitney U test or Kruskal–Wallis test were used to compare variables among groups, considering the numerousness of groups, for non-normally distributed variables. Normally distributed variables were compared through *t* test or ANOVA, according to the number of groups. Considering microbial taxa, phylum was expressed as % of total microbial population, whereas family, genus, species were expressed as OTUs (operational taxonomic units). Analysis of alpha diversity (within-sample) of gut microbiota composition, based on OTU data, was performed for the four groups of dogs. The alpha diversity of the gut microbiota was assessed with Shannon’s diversity index (reflecting both richness and evenness). Multivariate analysis was performed on the unweighted UniFrac distance matrixes using ANOSIM (Analysis of Similarity) test within PRIMER 7 software (PRIMER-E Ltd., Luton, UK) to analyze differences in phyla in different groups based on the presence of cholestasis, hyperlipemia, endocrine diseases, and different diets. Statistical significancy was identified for *p* < 0.05.

## 3. Results

### 3.1. Animals

In this prospective case-control study, 65 client-owned dogs with CHD were enrolled. According to breed, mongrels were more prevalent (*n* = 16) followed by Dachshund (*n* = 5), Cocker Spaniel (*n* = 5), Cavalier King Charles (*n* = 3), Golden Retriever (*n* = 3), Maltese (*n* = 3), West Highland White Terrier (*n* = 3), Toy Poodle (*n* = 3), Yorkshire (*n* = 3), Shih Tzu (*n* = 2), Chihuahua (*n* = 2), Jack Russell Terrier (*n* = 2), Pincher (*n* = 2), Labrador Retriever (*n* = 2), and one of each among Bull Terrier, Flat-coated Retriever, Breton, French Bulldog, Dobermann Pincher, English Setter, Boxer, German Shepherd, Apuan Shepherd, Pomeranian Spitz, and Zwergschnauzer.

The median age was 10 years (range 1.2–15.3), and sex was equally distributed with 32 female (20/32 neutered) and 33 male (6/33 neutered) dogs. Serum biochemical findings are reported in Table 1. Forty-three dogs (66%) out of 65 presented increased serum ALT. According to lipemia, 41 (63%) dogs had hyperlipemia, whereas the remaining 24 (37%) dogs had normal serum concentrations of both cholesterol and triglycerides.

Considering cholestasis, 45 (69%) dogs showed concurrent ultrasound and biochemical signs of cholestasis, and thus considered as presenting chronic cholestatic liver disease. Nineteen out of 65 (29%) dogs received diagnosis of concurrent endocrine disorder; in particular, diabetes mellitus (*n* = 6), hyperadrenocorticism (*n* = 4), hypothyroidism (*n* = 7), hypoadrenocorticism (*n* = 1), and Schmidt′s syndrome (*n* = 1). Twenty-six out of 65 (40%) dogs presented evident chronic gastrointestinal clinical signs (vomit, diarrhea) whereas the remaining 39 (60%) were asymptomatic. Only 18/65 (28%) of dogs had liver biopsies and histological examination at the moment of inclusion, in which 12 diagnosed with chronic hepatitis, 3 with hepatic vascular disorders (portal vein hypoplasia) and 3 with degenerative disorders. Considering diet, 5 (8%) dogs were feed with a diabetic diet, 9 (14%) with a hepatic diet, 19 (30%) with a gastrointestinal hyperdigestible diet, and the remaining 32 (48%) with a maintenance diet.

### 3.2. Microbiome Analysis

High-quality demultiplexed sequences were obtained from the fecal samples. Rarefaction analysis and alpha diversity measures showed that the bacterial communities were sufficiently sampled, and further sequencing would be unlikely to significantly increase the observed microbial diversity detected.

#### 3.2.1. Alpha Diversity

Alpha diversity analyzed using the Shannon index (reflecting both richness and evenness) and considering the subject and the group was 2.23 ± 0.67. Complete results are reported in Table 2.

#### 3.2.2. Multivariate Comparisons of Gut Microbiome Composition

Complete results of bacterial taxa in feces of CHD dogs are reported in Table 3.

At phylum level, the microbiota in all dogs was dominated by *Firmicutes* (mean 56.98%), *Proteobacteria* (mean 12.34%) *Bacteroidetes* (mean 4.72%), *Fusobacteria* (mean 3.13%), and *Actinobacteria* (mean 1.72%) (Figure 1).

Based on the ANOSIM analysis, no differences in phyla in ALT (*p* = 0.25), hyperlipemia (*p* = 0.43), endocrine disorders (*p* = 0.23), cholestasis (*p* = 0.9), and diet categories (*p* = 0.6) were found.

At family level, dogs with concurrent biochemical and ultrasonographic signs of cholestasis had a reduced level of *Bacteroidaceae*, *Enterobacteriaceae*, *Erysipelotrichaseae*, *Eubacteriaceae*, *Fusobacteriaceae*, and *Veillonellaceae* when compared to CHD dogs without evidence of cholestasis.

At the genus level, *Streptococcus* was significantly increased in dogs with increased serum ALT, whereas no statistically significant differences were observed for other evaluated taxa.

At the species level, dogs with hyperlipemia presented significant higher levels of *Salmonella enterica* and *Escherichia coli*, and lower levels of *Escherichia* and *Enterococcus faecium*. Dogs with endocrine diseases presented increased *Klebsiella* and *Prevotella* when compared to CHD dogs without endocrinopathy.

Dogs with concurrent biochemical and ultrasonographic signs of cholestasis presented significant higher levels of *Escherichia/Shigella* and *Serratia*, while *Clostridium hiranonis*, *Fusobacterium*, *Megamonas*, *Ruminococcus faecis*, *Escherichia*, *Streptococcus*, *Turicibacter* were less when compared to CHD dogs without evidence of cholestasis. *Lactobacilli* and *Bifidobacteria* were absent in all groups analyzed.

Significant differences in absolute OTU and percentage abundance of bacterial species in the four groups of dogs are presented in Figure 2 and Table 4. Complete results are reported in Table A1 and Table A2 (Appendix A).

Considering the diet category, most investigated bacterial taxa did not show a statistically significant difference, except for *Bacteroidetes*, *Helicobacter*, *Prevotella*, and *Ruminococcaceae*. Statistically significant differences are reported in Table 5. Complete results are reported in Table A3 (Appendix A).

## 4. Discussion

Among our population of CHD dogs, several GM modifications were observed, especially in the presence of cholestasis. Recently, the dysbiosis effect of medical management of CPSS [24] was investigated, with evidence of dysbiosis in relation to antibiotic/lactulose/antiacid therapy; however, no previous studies aimed to assess GM in a large cohort of CHD dogs. Hence, it is important to consider that the present results do not have a solid comparison in literature; thus, they must be considered as preliminary results. Comparison with the literature should be cautious, since it is well described that GM composition may deeply vary depending on factors such as sample storage method, analytic procedure, and taxonomic assignation [25], However, these findings clear the way for further investigations and suggest important clinical considerations.

The main phyla of bacteria present in the fecal microbiota of the dogs reported here is similar to what has been published previously: *Firmicutes*, *Proteobacteria*, *Fusobacteria*, and *Bacteroidetes* are the four most represented phyla [3]. In clinically healthy dogs and cats, *Firmicutes* usually represent the most common phyla, constituting about 45–70% of the total fecal microbiome [26,27]. High abundance of *Bacteroidetes* in the gut microbiota has been associated with the fecal concentration of short chain fatty acids (SCFAs), which modulate various metabolic functions [28]. Their presence in a healthy GM varies from 2% to 28% [26,27]. *Fusobacteria* constitute around 15% of total canine microbiome [26,27], and they seem to decrease in relation to chronic enteropathy [29]. The *Proteobacteria* and *Actinobacteria* phyla are typically colonizers of the small intestine and in healthy conditions will present in smaller numbers in fecal samples [26,27]. In clinically healthy adult dogs, relative abundances of *Proteobacteria* vary considerably and generally range from 0% to 22% [30]. *Proteobacteria* are frequently highlighted, as they include a number of clinically important gastrointestinal pathogens (diarrheogenic *Escherichia coli*, *Campylobacter jejuni*, *Klebsiella pneumoniae*, *Salmonella typhimurium*, *Yersenia enterocolitica*) [31]; however, they are also considered to contribute to amino acid, protein, and carbohydrate metabolism, and also maintain oxygen homeostasis in the gastrointestinal tract of healthy dogs [30].

Gut microbiome modification may arise from several mechanisms during CHD; however, deeper modification was observed in relation to cholestasis. Dogs who presented concurrent biochemical and ultrasonographic signs of cholestasis presented the most evident modifications, with increased *Escherichia shigella* and *Serratia*, and reduced *Bacteroidaceae*, *Clostridium (C. hiranonis)*, *Enterobacteriaceae*, *Erysipelotrichaceae*, *Eubacteriaceae*, *Fusobacteriaceae (Fusobacterium)*, *Megamonas*, *Ruminococcaceae* (*Ruminococcus*, *R. faecis*), *Escherichia*, *Streptococcus*, *Turicibacter*, and *Veillonellaceae*, when compared to CHD dogs without evidence of cholestasis. In dogs with cholestatic CHD, a dysbiotic pattern partially similar to that which develops during acute diarrhea was observed, specifically with a reduction in *Ruminococcaceae* and *Turicibacter* [29]. *Turicibacter* and *Ruminococcus* are associated with the metabolism of SCFAs, especially butyrate, involved in several physiological interactions between the GM and host. For this reason, they are currently considered beneficial, and have been shown to be less in dogs diagnosed with chronic enteropathy [17,32]. In humans, it is assumed that given the high SCFA concentration that reaches the liver through portal flow and its immune regulatory functions, SCFAs can influence immune tolerance and gut–liver immunity. Thus, SCFAs could be altered in immune liver diseases such as primary biliary cirrhosis, primary sclerosing cholangitis, and autoimmune hepatitis [33]. Conversely, the observed reduction in *Fusobacteriaceae* and *Bacteroidaceae*, both involved in SCFAs metabolism [34,35], is the pattern more frequently associated with chronic enteropathy (i.e., IBD) [36]. Similar to some studies in dogs with chronic inflammatory enteropathy, we observed increased *Escherichia/shigella*, a taxa that is anecdotally considered as an opportunistic pathogen [37]. *Escherichia/shigella* may be increased in relation to dysbiosis and dysregulation in eubiont/pathobiont interaction [38].

Interestingly, *Clostridium hiranonis*, which seems to play a major role in host health and maintaining a normal microbiota in dogs, was reduced in our population of dogs with cholestatic CHD [17,39]. *Clostridium hiranonis* is identified as extremely important in dogs since it seems to be directly involved in the production of secondary BAs from the deconjugation of primary BAs [17]. Hence, it represents one of the key microbes involved in the metabolism of BAs, and its reduction is related to increased primary BAs and decreased secondary BAs. In adequate amount, secondary BAs have been associated with immune regulation [40,41]. The alteration in bile quality and flow, typical of cholestasis, may have affected the levels of *Clostridium hiranonis*. In a vicious circle, is it also reasonable that lower levels of *Clostridium hiranonis* may have predisposed dogs to develop alteration in biliary physiology and cholestasis, affecting primary/secondary BAs proportion and thus BAs metabolism and the entero-hepatic circle. Similar to what we observed in relation to cholestasis in our population, *Megamonas* and *Veillonella* were found to be less in studies of the GM in human biliary tract diseases (i.e., PBC, PSC) [42,43]. They are involved in the metabolism of SCFAs, mostly acetate and propionate for *Veillonella* and butyrate for *Megamonas,* and thus related to human health [44]. SCFAs demonstrate several beneficial activities, such as contribute to maintaining the intestinal epithelial barrier functions and exhibiting immunomodulatory and anti-inflammatory properties [45]. Considering *Erysipelotrichaceae*, they were reduced if cholestasis was present. In humans and rats, they seem to be influenced by dietary fat, with decreased abundance in relation to low-fat diets [46,47]. In human cirrhotic patients, they seem to decrease in relation to body cell mass deficiency [48]. In hamsters, they showed a strong correlation to serum cholesterol [49]. However, their role during CHD and cholestatic CHD has not yet been extensively investigated [50]. We may suppose that cholestasis, through the alteration in biliary function, which encompasses lipid emulsion and digestion, could have influenced *Erysipelotrichaceae* abundance in our population.

In humans and dogs with CHD, several factors may influence dyslipidemia, such as diet (fat content and composition), cholestasis, and metabolic lipoprotein alteration in relation to hepatic injury [51,52]. In humans, dysbiosis is considered as both the cause and consequence of lipid metabolism alteration [53]. Endocrine disorders could be associated with dysregulation in lipid metabolism, resulting in hyperlipemia [52]; however, between our subgroup of dogs with/without hyperlipemia or endocrine disorders, different GM modifications were observed. This may suggest that many and different mechanisms may be involved in the influence of GM. Dogs with hyperlipemia presented significantly higher levels of *Salmonella (S. enterica)* and *Escherichia coli*, and lower levels of *Escherichia* and *Enterococcus faecium*. Higher levels of *Escherichia coli* could be related to dysbiosis, since it is considered to increase during dysbiosis processes [29].

In humans, a strong link between dysbiosis and endocrine disorders such as diabetes mellitus [54] and hypothyroidism [55,56] is assumed. Dysbiosis in GM composition and function is linked to immune-mediated and endocrine diseases through several mechanisms, such as predisposition to aberrant immunologic responses promoted by increased intestinal permeability and influence on the availability of essential micronutrients for the thyroid gland [56,57]. Additionally, several clinical trials showed benefits in the administration of probiotics in patients diagnosed with diabetes and hypothyroidism [58,59]. In the present study, individuals diagnosed with endocrine disease presented increased *Klebsiella* and *Prevotella* compared to dogs with primary liver disease. *Prevotella* was found to be higher in dogs diagnosed with endocrine disorder, including dogs with diabetes mellitus fed with a diabetic diet. In fact, *Prevotella* was also found to be higher in dogs fed with the diabetic diet; it could be related to its composition in carbohydrates and fiber since *Prevotella*, in both humans and dogs, seems to be strongly associated with a greater intake of a plant-based diet rich in fiber, simple sugars, and plant-derived compounds [60,61,62,63]. Thus, higher levels of *Prevotella* encountered in our population of dogs with endocrinopathy may be primarily affected by the high number of dogs fed the diabetic diet.

In our population, *Salmonella (S. enterica)* was increased in dogs with hyperlipemia, whereas *Klebsiella* was increased in dogs diagnosed with endocrine disorder. These taxa are currently considered as pathobionts that may overgrow and express their pathogenicity in presence of dysbiosis [64]. The gut microbiome influences gut barrier integrity by either maintaining an immune signaling mechanism or by producing metabolites like short chain fatty acids. Certainly, the presence of pathogenic bacteria can induce alterations in the epithelial barrier of the intestine. Compromised gut barrier integrity is likely to lead to translocation of microorganisms and microbe-derived molecules into the portal system. Under such a condition, these microbes and their biosynthesized metabolites can translocate to the liver from where they can be carried through the portal system to distal organs, thereby causing inflammation and injury. It is difficult to interpret the variation in pathogenic species in relation to CHD. According to the literature, the prevalence of *Salmonella* has been estimated to be around 2.5% in dogs, with almost half the salmonella-positive animals asymptomatic [65]. Thus, the pathogenic potential of *Salmonella* could be expressed in relation with other factors including dysbiosis and host–immune system interaction. Whereas, among humans, it was demonstrated that high-alcohol-producing *Klebsiella pneumoniae* could drive a condition of excess endogenous alcohol production associated with NAFLD [66]. The pathogenic role of these species is still an area of interest and further studies are needed to better understand their role in canine gastrointestinal health and disease.

*Streptococcus* was significantly increased in dogs with increased serum ALT. A recent study performed on human alcoholic liver disease (ALD) identified *Streptococcus*, as the predominant bacterium, positively correlated with AST level and considered a microbiological marker to evaluate the severity of liver injury in ALD patients [67]. Further studies are needed to evaluate whether *Streptococcus* may be associated with CHD severity in dogs.

Among the analyzed parameters, cholestasis seemed to strongly affect GM modifications. We can hypothesize that the alteration in bile flow and composition, typical of cholestasis, may directly affect the local intestinal microbial environment. Other alterations (increased ALT, hyperlipemia, endocrine disorder) and diet may affect GM in a more indirect way, and this could have resulted in less evident GM modification. For this reason, the presence of dysbiosis should be considered in dogs with CHD, especially in relation to cholestasis. These preliminary results suggest the presence of a gut–liver axis; future metabolomic assessment is important for better characterization. Further studies are needed to better understand the metabolic and clinical implications of GM alteration in dogs diagnosed with CHD and cholestatic CHD. This study must be considered in light of its limitations: first the fact that not all the CHD dogs underwent hepatic biopsy and histological characterization. Possible associations between canine GM composition and different types of CHDs should be investigated in the future. Moreover, a control group of healthy dogs was not recruited, and diet was not standardized among dogs due to ethical issues. Further studies with the concomitant evaluation of GM and metabolomic analytes, especially bile acids, in CHD dogs are needed to better investigate the gut–liver interaction. Lastly, it was decided to exclude dogs that received antibiotics during a two-month period before inclusion, but some GM modification caused by antibiotic administration can be detected after longer periods [68].

## 5. Conclusions

Gut microbial composition may be altered in dogs in relation to the presence of cholestasis, which seems to affect gut microorganisms, similar to human CHD. Clinicians may consider the presence of a gut–liver axis and dysbiosis in the management of dogs with cholestatic CHD, even if further studies are needed to point toward its metabolic and clinical implications.

## Figures and Tables

**Figure 1 animals-13-03174-f001:**
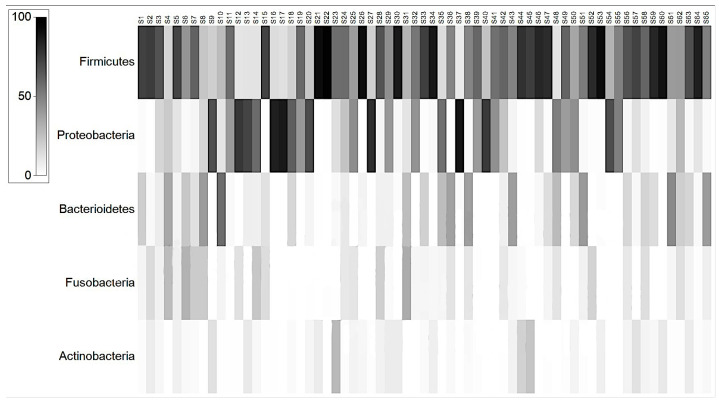
Shade graph representation of the microbiota phyla in the study population. The grayscale squares represent the % of each phyla in the total of microbiota species.

**Figure 2 animals-13-03174-f002:**
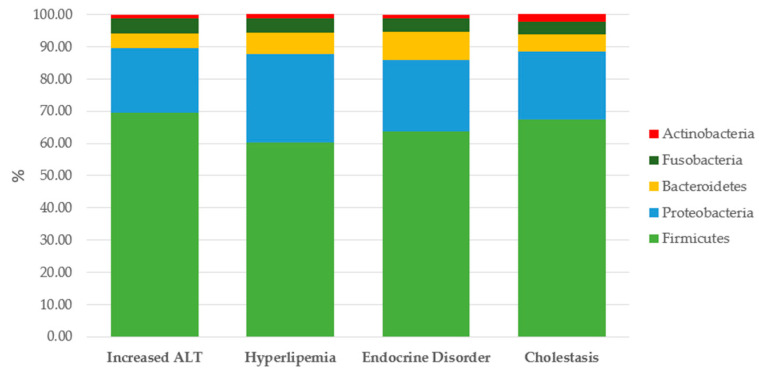
Abundance of main phyla in feces of dogs with CHD: comparison among different subgroups.

**Table 1 animals-13-03174-t001:** Descriptive statistics for serum hepatic enzymes, total bilirubin (Tot Bil), and cholesterol (Chol), expressed as median and range in CHD dogs. Data are expressed as medians and range.

	CHD Dogs	Reference Range
ALP	512 (43–8907)	45–250 U/L
GGT	3.4 (1.1–375)	2–11 U/L
AST	44 (18–1493)	15–40 U/L
ALT	113 (25–2652)	20–70 U/L
Tot Bil	0.22 (0.07–19.3)	0.07–0.3 mg/dL

**Table 2 animals-13-03174-t002:** Differences in alpha diversity (Shannon), number of species, in feces of CHD dogs considering serum ALT, evidence of hyperlipemia, endocrine disorder, or cholestasis, expressed as OTUs (operational taxonomic units). Data are expressed as means ± SD or medians and ranges. Applied statistical tests: unpaired *t* test, Kruskal–Wallis test.

	**ALT**			**Hyperlipemia**		
	**Normal (*n* = 22)**	**Increased (*n* = 43)**	** *p* ** **-Value**	**Present (*n* = 41)**	**Absent (*n* = 24)**	** *p* ** **-Value**
N° Species	17,223 ± 6270	16,640 ± 4208	0.65	16,737 ± 4205	17,008 ± 6135	0.83
Alpha diversity	2.217 ± 0.76	2.246 ± 0.62	0.87	2.190 ± 0.66	2.314 ± 0.68	0.47
	**Endocrine Disorder**			**Cholestasis**		
	**Present (*n* = 19)**	**Absent (*n* = 46)**	** *p* ** **-Value**	**Present (*n* = 45)**	**Absent (*n* = 20)**	** *p* ** **-Value**
N° Species	17,032 ± 5213	16,757 ± 4908	0.84	16,698 ± 4921	17,150 ± 5160	0.73
Alpha diversity	2.293 ± 0.74	2.212 ± 0.64	0.66	2.168 ± 0.64	2.388 ± 0.7	0.22
**Diet Category**						
	**Diabetic** **(*n* = 5)**	**Hepatic** **(*n* = 9)**		**Gastrointestinal** **(*n* = 19)**	**Maintenance** **(*n* = 32)**	** *p* ** **-Value**
N° Species	20,000 (10,000–29,000)	14,000 (8000–25,000)		14,000 (7000–23,000)	18,000 (11,500–30,000)	0.12
Alpha diversity	**3.02 (1.34–3.24)**	**1.57 (0.93–3.14)**		**2.3 (1.01–3.14)**	**2.54 (0.83–3.16)**	**0.02**

Bold text indicates the significant differences (*p*-Value < 0.05).

**Table 3 animals-13-03174-t003:** Medians and ranges of bacterial taxa in feces of CHD dogs. Phyla are expressed as % of total; family, genus, and species are expressed as OTUs.

Bacteria	Values
**Phylum**	
Actinobacteria	1.72 (0.01–27.29)
Bacteroidetes	4.72 (0.01–57.34)
Firmicutes	56.96 (8.72–98.39)
Fusobacteria	3.13 (0–33.29)
Proteobacteria	12.34 (0.02–89.13)
**Family**	
Bacteroidaceae	14 (0–60,949)
Bifidobacteriaceae	0 (0–1112)
Clostridiaceae	4774 (0–67,993)
Corynebacteriaceae	0 (0–136)
Enterobacteriaceae	14 (0–129,390)
Enterococcaceae	0 (0–63,111)
Erysipelotrichaceae	57 (0–82,064)
Eubacteriaceae	666 (0–72,100)
Fusobacteriaceae	23 (0–66,183)
Lactobacillaceae	0 (0–67,987)
Ruminococcaceae	47 (0–29,161)
Veillonellaceae	0 (0–102,369)
**Genus**	
Acinetobacter	0 (0–20)
Actinomyces	0 (0–672)
Bacteroides	0 (0–60,922)
Bifidobacterium	0 (0–1112)
Blautia	5328 (0–64,285)
Campylobacter	0 (0–21,646)
Clostridium	1685 (0–74,731)
Coprobacillus	0 (0–177)
Coprococcus	0 (0–477)
Corynebacterium	0 (0–34)
Cronobacter	0 (0–92)
Enterococcus	0 (0–62,340)
Escherichia	76 (0–8088)
Faecalibacterium	0 (0–25,117)
Fusobacterium	0 (0–58,851)
Helicobacter	0 (0–2234)
Klebsiella	0 (0–152)
Lactobacillus	0 (0–67,797)
Megamonas	0 (0–102,357)
Prevotella	42 (0–42,872)
Proteus	0 (0–415)
Pseudomonas	0 (0–734)
Ruminococcus	657 (0–40,941)
Salmonella	0 (0–796)
Serratia	0 (0–48,468)
Staphylococcus	0 (0–11)
Streptococcus	0 (0–84,431)
Turicibacter	0 (0–12,604)
Yersinia	0 (0–367)
**Species**	
*Clostridium difficile*	0 (0–8707)
*Clostridium hiranonis*	63 (0–7827)
*Clostridium perfringens*	642 (0–60,494)
*Enterococcus faecium*	0 (0–9053)
*Escherichia coli*	0 (0–392)
*Escherichia shigella*	0 (0–9108)
*Faecalibacterium prausnitzii*	0 (0–10,062)
*Helicobacter canis*	0 (0–218)
*Lactobacillus acidophilus*	0 (0–15,127)
*Ruminococcus faecis*	0 (0–5902)
*Salmonella enterica*	0 (0–30)

**Table 4 animals-13-03174-t004:** Significant differences in bacterial taxa in feces of CHD dogs considering serum alanine transaminase (ALT) the presence of hyperlipemia, cholestasis, and endocrine disease. Applied statistical test: Mann–Whitney U test.

	Increased ALT (*n* = 43)	*p*	Hyperlipemia (*n* = 41)	*p*	Endocrine Disorder (*n* = 19)	*p*	Cholestasis (*n* = 45)	*p*
Bacteroidaceae							↓	0.001
Enterobacteriaceae							↓	0.001
Erysipelotrichaceae							↓	0.0008
Eubacteriaceae							↓	0.0002
Fusobacteriaceae							↓	0.0115
Ruminococcaceae							↓	0.0001
Veillonellaceae							↓	0.001
Clostridium							↓	<0.0001
Escherichia			↓	0.02			↓	0.0002
Fusobacterium							↓	0.01
Helicobacter								
Klebsiella					↑	0.021		
Megamonas							↓	0.002
Prevotella					↑	0.022		
Ruminococcus							↓	<0.0001
Salmonella			↑	0.03				
Serratia							↑	0.005
Streptococcus	↑	0.03					↓	0.001
Turicibacter							↓	0.03
*Clostridium hiranonis*							↓	0.002
*Enterococcus faecium*			↓	0.03				
*Escherichia coli*			↑	0.02				
*Escherichia shigella*							↑	0.005
*Ruminococcus faecis*							↓	<0.0001
*Salmonella enterica*			↑	0.03				

**Table 5 animals-13-03174-t005:** Significant differences in bacterial taxa in feces of CHD dogs considering diet category. Phyla are expressed as % of the total; family, genus, and species are expressed as OTUs (operational taxonomic units). Data are expressed as means ±SD or medians and ranges. Applied statistical tests: ANOVA, Kruskal–Wallis test, pairwise comparison unpaired *t* test, Mann–Whitney U test.

	Gastointestinal (*n* = 19)	Maintenance (*n* = 32)	Diabetic (*n* = 5)	Hepatic (*n* = 9)	*p*
Alpha Diversity (Shannon)	2.3 (1.01–3.14) ^d^	2.54 (0.83–3.16) ^d^	3.02 (1.34–3.24) ^d^	1.57 (0.93–3.14) ^a,b,c^	0.02
Ruminococcaceae	1749 (0–22,925) ^d^	1066 (13–26,808)	455 (68–29,161)	182 (0–638) ^a^	0.03
Helicobacter	0 (0–780) ^c^	0 (0–2234)	222 (0–1071) ^a,d^	0 (0–123) ^c^	0.03
Prevotella	0 (0–36,829) ^c^	0 (0–60,858)	653 (0–39,549) ^a,d^	0 (0–122) ^c^	0.04

^a^ Statistically different from gastrointestinal; ^b^ statistically different from maintenance; ^c^ statistically different from diabetic; ^d^ statistically different from hepatic.

## Data Availability

The complete dataset is available upon reasonable request.

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
