# Peer review of "Intestinal Microbiome in Dogs with Chronic Hepatobiliary Disease: Can We Talk about the Gut–Liver Axis?"

_animals, 2023, doi:10.3390/ani13203174_

Round 1
Reviewer 1 Report
Review Intestinal microbiome in dogs with chronic liver disease: can we talk about Gut-Liver axis?.
This study investigates differences in microbial fecal population of 65 dogs with chronic liver disease and other indicators of health: elevated ALT, hyperlipidemia, endocrine disorder, and cholestasis.
It is limited by the fact that diet is variable and there are no healthy pets to use as an anchor for comparison.
There is an error in the statistical handling of data. The phylum analysis as I understand it from the description was to evaluate the outcome variable of percentages using ANOVA. This is not correct as the compositional data should be analyzed in a way that accounts for this interdependence of response variables. See - Fernandes, A.D., Reid, J.N., Macklaim, J.M., McMurrough, T.A., Edgell, D.R. and Gloor, G.B., 2014. Unifying the analysis of high-throughput sequencing datasets: characterizing RNA-seq, 16S rRNA gene sequencing and selective growth experiments by compositional data analysis. Microbiome, 2, pp.1-13.
Line 115 the wording is confusing. Do you mean – “The Ion 16S™ Metagenomics Kit was used for library construction” please clarify.
Line 157 the authors state “Serum biochemical findings of are reported” this should be rewritten for clarity as it seems “of” is a typo.
Line 179 (median, mean) actually seems like a typo, as written it has no discernable meaning.
Table 2 mean separation is not reported for the alpha diversity change associated with food fed. These have very unequal n and the very real possibility that the assumption of equal variance is incorrect. I assume the reported numbers are means with the overall variance used with the number of dogs per category to calculate standard errors. Minimally this table needs to describe what is reported and how mean separation is done as well as what means actually are different. Also, the methods for assuring equal variance within treatment is needed. Also, for completion add the number of dogs per category for all comparisons (now done for all excepting the diet category where it should be added). If thesea changes are included in Table 2 then it’s not needed to be repeated in Table 5.
Figure 2 – these plots were not easy to interpret as the 3rd axis was completely lost in the presentation. I find them unacceptable, however there may be a journal standard, please attempt to improve the presentation.
Table 3. This needs some information: what is actually presented? I sort of hope it’s showing the median and range yet it’s unknown. This needs to be described.
Line 432
I note the typo” Authors should discuss the results and how they can be interpreted from the perspective of previous studies and of the working hypotheses. The findings and their implications should be discussed in the broadest context possible. Future research directions may also be highlighted.” This should be removed.
Conclusions –
These are not strictly based on the data presented and must be rewritten. Basically, the conclusion “Gut microbial composition may be altered in dogs with CLD” cannot be inferred by this study because no healthy controls are used. What is clear is that dogs with CLD and cholestasis have an altered gut microbial population as compared to dogs with CLD. This leads to the discussion about the gut-liver axis which as stated in the title we can discuss but this experiment does not quantify.
In summary, this is an interesting paper and as in much good science leads to interesting discussion and the need for more experimentation. However, the correct statistical analysis must be done and a number of typographical errors and rhetorical overreach must be corrected. Because of this I have stated there is a need for a "major" revision. This is not so much tear it down and start again but changes required that may make a major change in the paper.
Minor adjustments needed.
Author Response
Review Intestinal microbiome in dogs with chronic liver disease: can we talk about Gut-Liver axis?.
This study investigates differences in microbial fecal population of 65 dogs with chronic liver disease and other indicators of health: elevated ALT, hyperlipidemia, endocrine disorder, and cholestasis.
It is limited by the fact that diet is variable and there are no healthy pets to use as an anchor for comparison.
Authors: Absolutely they represent a limit, we added the fact that a control group is missing in the sentence: “Moreover, a control group of healthy dogs was not recruited and, due to ethical issues, diet was not standardized among dogs”.
There is an error in the statistical handling of data. The phylum analysis as I understand it from the description was to evaluate the outcome variable of percentages using ANOVA. This is not correct as the compositional data should be analyzed in a way that accounts for this interdependence of response variables. See - Fernandes, A.D., Reid, J.N., Macklaim, J.M., McMurrough, T.A., Edgell, D.R. and Gloor, G.B., 2014. Unifying the analysis of high-throughput sequencing datasets: characterizing RNA-seq, 16S rRNA gene sequencing and selective growth experiments by compositional data analysis. Microbiome, 2, pp.1-13.
Authors: Dear Reviewer, thank you for your suggestion. We modified the statistical analysis, considering others similar studies, analyzing phyla with ANOSIM analysis and modified the manuscript accordingly. We only had few significant differences and we lost significancy applying this test (Stastistical analysis, results and discussions were modified)
Line 115 the wording is confusing. Do you mean – “The Ion 16S™ Metagenomics Kit was used for library construction” please clarify.
Authors: Thank you for your suggestion, that was an error in the sentence construction. We modified the text accordingly.
Line 157 the authors state “Serum biochemical findings of are reported” this should be rewritten for clarity as it seems “of” is a typo.
Authors: Thank you for your suggestion, that was definitely a typo. We modified the text accordingly.
Line 179 (median, mean) actually seems like a typo, as written it has no discernable meaning.
Authors: Thank you for your suggestion, that was definitely a typo. We modified the text accordingly.
Table 2 mean separation is not reported for the alpha diversity change associated with food fed. These have very unequal n and the very real possibility that the assumption of equal variance is incorrect. I assume the reported numbers are means with the overall variance used with the number of dogs per category to calculate standard errors. Minimally this table needs to describe what is reported and how mean separation is done as well as what means actually are different. Also, the methods for assuring equal variance within treatment is needed. Also, for completion add the number of dogs per category for all comparisons (now done for all excepting the diet category where it should be added). If the sea changes are included in Table 2 then it’s not needed to be repeated in Table 5.
Authors: Thank you for your suggestion, considering very different population among diet category, we applied Kruskall-Wallis test and thus, express data as medians and ranges. We performed all the other changes required.
Figure 2 – these plots were not easy to interpret as the 3rd axis was completely lost in the presentation. I find them unacceptable, however there may be a journal standard, please attempt to improve the presentation.
Authors: Dear reviewer, thank you for your observation. Since those graphs, representing beta-diversity, are not statistically analyzed and thus scarcely interpretable, we decided to remove them from the manuscript.
Table 3. This needs some information: what is actually presented? I sort of hope it’s showing the median and range yet it’s unknown. This needs to be described.
Authors: Dear reviewer, thank you for your comment, we actually mean that medians and range are reported. We modified the text accordingly and also noticed that other tables had the same error (Table 1, 2, 3, 5 and A, B, C).
Line 432
I note the typo” Authors should discuss the results and how they can be interpreted from the perspective of previous studies and of the working hypotheses. The findings and their implications should be discussed in the broadest context possible. Future research directions may also be highlighted.” This should be removed.
Authors: Thank you very much for your observation, that was a typo. We modified the text accordingly.
Conclusions –
These are not strictly based on the data presented and must be rewritten. Basically, the conclusion “Gut microbial composition may be altered in dogs with CLD” cannot be inferred by this study because no healthy controls are used. What is clear is that dogs with CLD and cholestasis have an altered gut microbial population as compared to dogs with CLD. This leads to the discussion about the gut-liver axis which as stated in the title we can discuss but this experiment does not quantify.
Authors: Thank you, we agree with your comment. Conclusions should be reduced to what was strictly evaluated, we modified the text accordingly.
In summary, this is an interesting paper and as in much good science leads to interesting discussion and the need for more experimentation. However, the correct statistical analysis must be done and a number of typographical errors and rhetorical overreach must be corrected. Because of this I have stated there is a need for a "major" revision. This is not so much tear it down and start again but changes required that may make a major change in the paper.
Reviewer 2 Report
The authors describe a study where the gut microbiome was analyzed in dogs with elevated liver enzymes. The study showed that several bacterial modifications were observed, especially with cholestasis. Dysbiosis can be seen in dogs with a cholestatic pattern for elevated liver enzymes.
My biggest concern with this paper is that the definition of chronic liver disease in small animal medicine is unacceptable. Medical history, physical examination, hematology, blood biochemistry, abdominal ultrasonography was what were used to determine chronic liver disease which is not an acceptable definition of CLD. Unless there is histopathology, one cannot establish CLD. So to make sense of this data, one option would be to change CLD to persisting serial elevations in liver enzymes or to focus on cholestatic disease based on enzymes and changes seen on ultrasound. This is the backbone of the paper.
This is acceptable
Author Response
The authors describe a study where the gut microbiome was analyzed in dogs with elevated liver enzymes. The study showed that several bacterial modifications were observed, especially with cholestasis. Dysbiosis can be seen in dogs with a cholestatic pattern for elevated liver enzymes.
My biggest concern with this paper is that the definition of chronic liver disease in small animal medicine is unacceptable. Medical history, physical examination, hematology, blood biochemistry, abdominal ultrasonography was what were used to determine chronic liver disease which is not an acceptable definition of CLD. Unless there is histopathology, one cannot establish CLD. So to make sense of this data, one option would be to change CLD to persisting serial elevations in liver enzymes or to focus on cholestatic disease based on enzymes and changes seen on ultrasound. This is the backbone of the paper.
Authors: Dear reviewer, thank you for your comment, we hope we fully understood your point of view. Our aim was to describe gut microbiome in dogs with various kink of chronic hepatobiliary disease, but we understand that the employed term Chronic Liver Disease, could be confusing for the reader.
We did not refer to Chronic Hepatitis (histopathology required for diagnosis), which were included in our population but did not represent the only category of chronic liver diseases we were interested to evaluate.
For sure, hystopathology could be useful to better characterize the hepatic pathologic process (we clearly state it in our limits), but for our aim, we believe that having persisting altered enzymes and persisting ultrasonographic liver/biliary tract modifications could be enough to point out the presence of a Chronic liver disease.
We can modify the term “Chronic Liver Diseases” which can be mistaken for “Chronic Hepatitis” into the more correct term “Chronic Hepatobiliary Diseases”, we modified the text accordingly.
We decided to evaluate both enzymes and ultrasonographic modification considering that some chronic liver diseases (vascular abnormalities, pre-cirrhotic liver) could be present also in the absence of increased liver enzymes. And lastly, we decided to include both dogs with and without sings of cholestasis, to have the possibility to compare them between each other evaluating the potential impact of cholestasis on gut microbiome.
Round 2
Reviewer 1 Report
I appreciate your responses and believe the paper is now ready for publication with one small change. The statement from "122.633 to 233946 per sample" seems likely to be in error (the "." being hard to interpret). This should be changed.
Author Response
Reviewer: I appreciate your responses and believe the paper is now ready for publication with one small change. The statement from "122.633 to 233946 per sample" seems likely to be in error (the "." being hard to interpret). This should be changed.
Authors: Dear reviewer, thank you very much for your previous revisions, we really think they improved the quality of our work. We modified the text according to this last comment. The authors removed the sentence “with the number of reads ranging from 122.633 to 233946 per sample” because it was not relevant.